# Effect of Applied Stress on T91 Steel Performance in Liquid Lead at 400 °C

**DOI:** 10.3390/ma11122512

**Published:** 2018-12-11

**Authors:** Anna Hojná, Fosca Di Gabriele, Michal Chocholoušek, Lucia Rozumová, Jan Vít

**Affiliations:** Centrum vyzkumu Rez (CVR), Rez 130, 250 68 Husinec, Czech Republic; fosca.di_gabriele@cvrez.cz (F.D.G.); michal.chocholousek@cvrez.cz (M.C.); lucia.rozumova@cvrez.cz (L.R.); jan.vit@cvrez.cz (J.V.)

**Keywords:** liquid metal embrittlement, ferritic-martensitic steel, T91, liquid lead, bend specimen, tapered specimen, oxidation

## Abstract

The environmental performance of structural materials (e.g., corrosion and environmental assisted cracking) is one of the critical issues being addressed in the development of Generation IV fast reactors. This work aims to support the study of the performance of the ferritic-martensitic steel T91 in liquid lead, under close to operation conditions, in order to assess its suitability for an application in lead fast reactors. T91 steel was tested in air and liquid lead at 400 °C using static and slow loading regimes. Applied stresses were chosen to be slightly above the yield strength in order to evaluate the threshold stress and strain for crack initiation. Three-point bending static exposure and constant extension tests of tapered specimens were performed. Post-test surfaces and cross sections of specimens were observed using scanning electron microscopy techniques in order to detect cracks and to analyze oxide layers. The effect of strain rate of the oxide layer cracking was observed. In conclusion, T91 was not susceptible to liquid metal embrittlement, a special case of environmentally assisted cracking under the testing conditions. The cracking conditions are discussed based on previous experience.

## 1. Introduction

The choice of structural materials based on their compatibility with the coolant is one of the most critical issues in the development of lead-cooled generation IV fast reactors (LFRs). T91, the 9% Cr ferritic-martensitic steel, is one potential material candidate for LFR. The compatibility of materials under LFR operational conditions needs to be investigated by means of corrosion-mechanical testing under selected conditions. These conditions should include a liquid lead environment, with specific oxygen content at temperatures ranging from 380 to 550 °C and application of stress up to the level allowed in operational design [1].

The performance of T91 in heavy liquid metals (HLMs) under the applied stress has been considered in most of the research studies on Pb-Bi eutectic (LBE) [2,3,4,5,6,7,8,9,10,11], while less attention has been paid to pure Pb [12,13,14]. Although one of the HLM coolant operation strategies is to maintain a specific oxygen content in liquid metals to build protective oxides on the steel, these media (Pb and LBE) do not have the same effect on materials. Owing to the different chemistry and range of service temperatures, different oxygen content is required to prevent environmental degradation such as dissolution and liquid metal embrittlement (LME). For instance, an optimal oxygen concentration to grow stable and protective oxide scales in liquid LBE and Pb was calculated to about one range, this being around 10^−7^ wt. % at 400 °C and 10^−6^ wt. % at 500 °C [15]. The narrow stable oxide window requires precise control of the oxygen content in the system operation. Corrosion testing of T91 steel in LBE, with the above oxygen range, has confirmed the oxygen protection [16,17,18]. However, corrosion testing continues in liquid Pb.

It has been recognized that LME should be considered as a special case in regards to environmentally assisted cracking (EAC), and that the wetting of material by liquid metal is an LME pre-condition. The LME of T91 steel occurs in LBE at temperatures up to 350 °C, that is, to about 220 °C above the melting point (124 °C). In particular, cases of LME susceptibility have been reported for T91 (supplied by different producers) in LBE, with no specified oxygen contents, at 160–350 °C [8,9] and in Pb at 350–400 °C [12,13,14]. Thus, the effect of the applied stress on T91 performance is now in need of research, so as to study in liquid Pb in the lower temperature range relevant to LFR operation.

In previous research [13,14], the susceptibility of T91 steel to LME in liquid Pb (oxygen content from 1 × 10^−11^ to 1 × 10^−7^ wt. %) was investigated by applying slow loading, up to rupture. In particular, slow strain rate tests of smooth and notched round bars were performed in a wide range of strain rates (from 10^−2^ to 10^−6^ s^−1^). Surface pre-treatment was applied to stimulate Pb wetting. The LME crack initiation and growth were only induced owing to multi-axial stresses and high local plastic strains, located close to the onset of a ductile void creation, in the notched specimens in the temperature range of 350–400 °C. The LME severity—the percentage of fracture surface covered by cleavage-like fracture—was the highest at 350 °C and decreased with increasing temperature. No LME occurred at 450 °C.

However, in previous studies LME initiation and growth has appeared only at high stress–strain conditions, which are beyond the operational conditions of the LFR design, and are similar to accident conditions. It has not yet been verified whether the LME crack can be initiated in less-severe conditions during long-term exposure. It is important for LFR designers to understand if wetting can occur locally under static stress in long-term exposure to liquid Pb at the oxygen level proposed for operation. Another question to answer is whether a crack can be initiated at lower stress near the yielding point without evident plastic strain contribution. Therefore, in this paper, T91 performance in liquid lead is studied predominantly with applied stresses around the yield strength in static and quasi-static load regimes. Moreover, several experiments are performed to evaluate threshold stress and strain for LME initiation.

## 2. Materials and Methods

### 2.1. Material

The ferritic-martensitic steel T91 (Grade 91 Class 2/S50460) of nominal composition given in Table 1 was produced by Industeel, ArcelorMittal group. The material was normalized at 1150 °C for 15 min with subsequent water cooling to room temperature and finally annealed at 770 °C for 45 min, then slowly cooled in the air. A typical microstructure formed by this heat treatment consists of lath martensite in original austenitic grains with an average grain size of 20 μm [14]. Mechanical properties in air at room and elevated temperatures are given in Table 2.

### 2.2. Specimens

Two types of specimens were applied—flat tapered tensile specimens and bend beams. A flat tapered specimen (Figure 1) with a thickness of 3 mm, width from 4 mm in the narrowest area to 6.4 mm in the widest one, and a gauge length of 26 mm was designed. The 3° taper creates a variation of stress and strain along the gauge length during mechanical testing. Maximum stress is always achieved in the minimum of the cross section; the stress level in the area close to the end of the widest part stays elastic and does not overcome the yield strength. This allows the identification of threshold stress and strain conditions for the crack initiation within a single test specimen. The specimens were fabricated by wire cutting using an electrical discharge machine (EDM). After machining, one of the two flat parallel surfaces was polished to a 1 µm finish (Figure 1b) while the other one was ground using a final 500-grid finish in the direction tilted 25 ± 10° to the load axis (Figure 1c). The specimen sides were not treated. Secondary particles appeared on the polished surface, while flaps and machining grooves dominated on the ground one. The surface roughness was measured along the specimen’s longitudinal axis using a DektakXT stylus profiler (Bruker). The arithmetical mean roughness (Ra) of the polished and ground surfaces were 0.005 µm and 0.047 µm, respectively. The scan length was 20 mm and evaluation length—0.25 mm. The hardness values of the polished and ground surfaces were 226 ± 5 HV30 and 230 ± 5 HV30, respectively.

Bend specimens (Figure 2a) of 1 mm thickness, 5 mm width, and 22 mm length were fabricated by EDM. The surface was ground to a 500-grid finish (Figure 2b). The surface roughness was measured along the specimen’s longitudinal axis by means of a DektakXT stylus profiler (Bruker). The Ra of the bend specimen was 0.249 µm (scan length 18 mm, evaluation length 0.25 mm). The hardness of the original surface was measured, and the value was 228 ± 5 HV30.

### 2.3. Experiment

#### 2.3.1. Constant Extension Rate Tensile Test

In the constant extension rate tensile (CERT) testing the tapered specimen was monotonically loaded in displacement control mode. Three displacement rates (R2 < R1 < R0) were applied ranging from 2 × 10^−8^ to 2 × 10^−4^ m/s.

The testing at 400 °C in air was performed using the electromechanical testing machine Z250 (Zwick/Roell). Tests at 400 °C in liquid Pb were performed in the HLM system built on the Kappa 50DS electromechanical creep testing machine (Zwick/Roell). The HLM system is based on the two-vessel concept, where the first vessel serves for the preparation of heavy liquid metals (melting and oxygen setting by Ar/H_2_ gas bubbling) and the second one for the testing of a specimen. The oxygen content was measured in both vessels using oxygen sensors (ref. Bi/Bi_2_O_3_). Each specimen was fixed into grips inside the second vessel then filled with liquid Pb which was melted in the first one.

#### 2.3.2. Three-Point Bend Exposure Test

Bend specimens were mounted into a special holder enabling pre-loading of four three-specimen sets to four different load levels (Figure 2a). Particularly, three pre-stress levels were applied (i.e., 80, 100, and 110% of Rp0.2, corresponding to specimens T081–T083, T101–T103, T111–T113, respectively). The fourth set was left without loading (i.e., T001–T003). The three-specimen sets were pre-bent by screwing up to required deflections at the test temperature according to the ISO 7539-2 standard equation:y = σ H^2^/(6Et),(1)
where σ is the applied stress, H is the span, E is Young’s modulus, and t is the specimen thickness. The holder was then inserted into the air-tight chamber of the static tank instrumented with the gas system. The static tank consisted of a stainless-steel autoclave with an inner alumina (Al_2_O_3_) crucible which prevented direct contact of the steel walls with the Pb during testing. In the sealed autoclave, the Pb was melted and purged with Ar+H_2_ (6%) gas to reduce the initial oxygen content to 10^−7^ wt. % at 400 °C. Oxygen sensors based on the Bi/Bi_2_O_3_ reference electrode were used. Only after the oxygen concentration became stable were samples inserted into the Pb bath from the air-tight chamber. Active oxygen control (i.e., maintaining the oxygen concentration in the Pb at the targeted level) was ensured by means of the automatic mixing of cover gases (Ar+H_2_ (6%) with Ar+O_2_ (10%)) controlled by the oxygen sensor signal.

### 2.4. Post-Test Evaluation

After exposure, bend and tapered specimens were first observed by microscopic techniques for crack occurrence. Then, cross and longitudinal section samples were prepared for investigation. If remaining lead obscured the observation, the specimens were chemically cleaned by immersion in a solution of H_2_O_2_, CH_3_COOH, and CH_3_CH_2_OH (1:1:1). The specimen surface was observed and analyzed using a MIRA3 GMU SEM and a dual beam FIB-SEM system LYRA3 GMU (TESCAN, Brno, Czech Republic). The surfaces were analyzed in secondary electron (SE) and back-scattering electron (BSE) modes at accelerating voltage of 15–20 kV.

## 3. Results

### 3.1. Bend Specimens

The exposure of pre-stressed specimens in liquid Pb at 400 °C, with an oxygen content of 10^−7^ wt. %, lasted for 1000 h. A total of twelve specimens were exposed: three of them were pre-loaded up to 110% of the yield strength, three up to 100%, three up to 80%, and three were tested without pre-loading.

During the initial 100 h, the oxygen concentration was kept close to the target value, as shown in Figure 3a. Then, for about 200 h, the oxygen concentration was optimized by dosing with an Ar+O_2_ (10%) and pure Ar gas mix. However, the measured oxygen level in the static tank was above the target value to feed the oxidation process at the specimen/holder surface. After that, the oxygen concentration was restored to the targeted 10^−7^ wt. % level (i.e., from 340 to 1000 h).

#### 3.1.1. Surface

After the exposure to liquid lead, the surfaces of bend specimens were covered with a continuous layer of lead. To observe the specimen surface, this layer was removed chemically.

The surface after the 1000 h exposure and the chemical cleaning of Pb showed areas with non-continuous and continuous oxide layer distribution (Figure 3b,c). No visible differences between the ground and polished surfaces of specimens without and with applied load were observed using SEM. Besides several imperfections in the oxide layer (Figure 3c), no initiated cracks were observed.

#### 3.1.2. Cross Section

Longitudinal cross sections of selected bend specimens were prepared for SEM observation. It was shown that distinctive oxide layers were built on the surface (Figure 4a,b). The thickness of the oxide layer ranged from about 1.5 to 5 µm. The oxide thickness was found to be in this range for both non-pre-stressed and pre-stressed specimens. 

The oxide double layer structure was observed using SE and BSE and confirmed by EDS line scans (Figure 4c,d). The outer oxide layer was a Fe-O-rich oxide and the inner one consisted of a Fe-Cr-O-rich phase. In several areas, the interface between the inner layer and the bulk was very uneven, whether a pre-stress was applied or not. Slight protuberances towards the bulk were observed at the interface. It is likely that the oxidation was accelerated in these areas, which might correspond to microstructural elements of about 0.2–0.5 µm size (i.e., much smaller than the grain size). 

### 3.2. Tapered Specimens

CERT testing of tapered specimens was performed at 400 °C in air and in Pb, with an oxygen content of 1 × 10^−6^ wt. %, using three displacement rates: R0 = 2 × 10^−4^, R1 = 2 × 10^−6^, and R2 = 2 × 10^−8^ m/s. Four specimens were tested in air and three in the Pb (Table 3). The test curves—the engineering stress in the minimum cross section vs. displacement—are shown in Figure 5. The first specimen was loaded in air to rupture to distinguish the target load for upcoming tests, which were to be stopped around the maximum load. The maximum stresses and corresponding displacements exhibited a scatter of about 10%, which is typical for the specimen-to-specimen scatter of steels. The maximum stress achieved in Pb was about 10% lower than the maximum one of the four specimens tested in air. The stress–displacement curves of tests with applied test rates R0 and R1 were smooth in air as well as in Pb, but both curves with R2 were non-uniform. The non-uniform behavior might be due to the material properties, since two different electromechanical test machines were used for the tests in air and in Pb—however, the effect of Pb on the non-uniform behavior was excluded.

The time dependence of temperature and oxygen concentration of three specimens tested in liquid Pb (T36, T37, and T38) are given in Figure 6. Zero time was set to the moment when liquid Pb was transferred to the second (test) tank. The vertical line marks the CERT starting point. The total time of the three CERT tests varied because of three different applied rates. The graphs show that the course of oxygen concentration in the second tank was similar for T36 and T37, but different for T38. After the filling, the oxygen concentration increased for about 2 h, and then decreased to the requested level within 5 h. On the other hand, the oxygen content did not increase but decreased to the level in about 5 h before the testing of T38. In spite of that, the loading of each test was begun after the stabilization of the oxygen concentration in the test tank. The history of oxygen concentration likely had a large impact on the initial oxide layer development. This is discussed in more detail in the following chapter.

Table 3 shows the results of the tests on tapered specimens. Maximum stress (σ_max_) and strain (ε_max_) values were calculated for the minimal cross section. σ_max_ equals to the maximum measured load divided by the original cross section, S_min_, before test. ε_max_ was estimated from the plastic part of total displacement (f_pl_) and the gauge length (L), ε_max_ = f_pl_/L × 100. Further, the results were also evaluated based on the microscopy observation using the procedure described in Section 3.2.1.

#### 3.2.1. Surface

After the CERT testing in air, the surfaces of tapered specimens were covered by oxides. Plastic deformation features (slip bands and shear) were well-formed and visible on the surface of specimen T29 loaded to rupture. The developed oxide was thick and homogeneous, such that it almost covered up the surface topographic inhomogeneities and irregularities. The only indications of the deformation appeared on the surface of the specimens loaded up to the maximum load. However, they were not well recognizable owing to the oxides obscuring fine features (see Figure 7a). Moreover, several ductile dimples were observed around inclusions on the polished surface and machining grooves on the ground one. No strain rate effect on the surface appearance was detected.

After the CERT testing in Pb, the surfaces of the tapered specimens were covered by non-uniform residuals of lead, and chemical cleaning was necessary in order to observe surface features.

The surface state was similar for specimens T36 and T37. On the polished surface, superficial cracks and indications of plastic deformation under the oxide layer were found in the minimum cross section area (see Figure 7b). On the ground surface, the machining grooves were still visible before the cleaning (Figure 8a,b). Figure 8a,c shows plenty of cracks in the oxide oriented perpendicularly to the load axis in the minimum cross section. Moreover, the oxide layer was locally gapped in several areas, where it was connected to the superficial cracks and the deformation bands (Figure 8c).

The number of the surface oxide cracks decreased towards the wider end of the tapered specimen, and at some point no cracks were observed. The furthest site from the minimum cross section, where the crack was observed, is the last crack distance (L_x_). Figure 8d shows the crack occurring at L_x_, where the deformation of the oxide layer was not visible. Threshold stress (σ_th_) for cracking in the test system was calculated from the width of the tapered specimen measured at L_x_ and the maximum stress applied during the test. Plastic strain, ε_Lx_, at L_x_ was estimated assuming a linear strain distribution along the gauge length from 0.2% strain in the position of the yield strength and ε_max_ in the minimal cross section. σ_th_ and ε_Lx_ values of each specimen are also given in Table 3.

In the case of specimen T38, the superficial cracks were scattered. Figure 9 shows the appearance of the ground and polished surfaces in the minimum cross section area, after chemical cleaning. The porous outer oxide layer remained on the surface without any massive spalling. Several cracks were observed, particularly in regions with the outer layer burnt out or removed by the cleaning (i.e., close to the edge in Figure 9b).

No effects of the ground or polished surfaces on the oxide cracking were observed in the specimens.

#### 3.2.2. Cross Section

The longitudinal cross section of selected tapered specimens was prepared for SEM observation. It was shown that the surface was covered by a continuous double layer oxide (Figure 10 and Figure 11). The outer oxide structure was Fe-O and the inner one consisted of Cr-Fe-O.

In the case of specimens T36 and T37, the oxide thickness was about 1.4–2 µm. The oxide was damaged in a number of sites. Multiple cracks going from the surface across the oxide and the detachment of two oxide layers were also observed in several sites. However, the depths of surface cracks were limited to the oxide width (Figure 10a). All cracks stopped at the interface between the inner oxide and the bulk. No crack was found to continue into the bulk of the metal.

On the other hand, the observation of the cross section of specimen T38 confirmed the previous statement that no massive cracking occurred. Several defects were detected in the outer layer, and two oxide layers were detached in a number of sites (Figure 11a,b). The oxide thickness was about 1.5–2.2 µm. SEM SE, BSE, and EDS observations showed that the oxide on the T38 specimen could not be distinguished from the other by any features.

## 4. Discussion

### 4.1. Experiments under Static and Quasi-Static Conditions

These experiments were performed to verify if the LME crack could be initiated under similar-to-design normal operational conditions for T91 steel in contact with liquid Pb. Three-point bend static and CERT quasi-static load testing in a simulated LFR environment were performed—namely, in liquid Pb at 400 °C assuming the oxygen content of 10^−7^ wt. %. The conditions were applied for up to 1000 h. For all tested T91 specimens, no LME crack initiation was detected. This can be explained by either the fact that wetting of T91 by liquid Pb did not occur or due to the loading (strain rate, maximum stress and strain) conditions, which might be unsuitable.

Post-observations after the static tests showed that oxides of 2–5 µm were grown on the surface. They were further analyzed by EDS and were identified as an outer Fe_3_O_4_ and inner spinel-type Fe-Cr-O oxides. This is also in agreement with previous studies [19,20,21], where these oxides were defined as highly protective. For all the bend specimens observed, no trend of breaking through the oxide layer within the range of applied static stresses was established. Only slight changes in the oxide thickness and some insignificant inner extensions were observed. The thickness was slightly lower for non-pre-stressed specimens in comparison to the pre-stressed ones. At the bulk–oxide interface, minor extensions of the spinel oxide towards the bulk material (Figure 4b) might be connected to the presence of stresses and the subsequent formation of inner easy-diffusion paths. This might also indicate the initiation stage of the development of the internal oxidation zone (IOZ) [20]. These areas might behave as local micro-notches, from which LME could initiate if a high load is applied (e.g., resulting from out-of-operational conditions). In CERT testing, the oxide scale was built for a shorter time (in the range of 5 to 50 h, much less than 1000 h) and was thinner than during the static tests. It was shown that the oxide scale on T36 and T37 specimens was built at slightly higher oxygen content (1 × 10^−6^ wt. %) including history of development than the scale on specimen T38 (8 × 10^−7^ wt. %), nevertheless, it was less resistant to the loading. Plenty of cracks between two layers, as well as across the scales in the sites loaded above the threshold stress and strain conditions, were detected in T36 and T37 oxide scales. However, the cracks did not go through the whole oxide scale. This can be connected with the quicker oxidation of crack tip opening, so a path for liquid lead to wet the metal was not created. On the other hand, mainly the cracking between the two layers and very rarely across the scale was observed in T38. Even if small differences in morphology of the oxide layers were observed, the oxides had same chemistry. It is likely that it cannot fully explain the different resistances.

The second reason for the fact that LME was not observed could be the loading (i.e., strain rate, maximum load) conditions, which might not be suitable to stimulate crack initiation and propagation. Naturally, the experiments were designed outside the already-proven LME conditions, to verify whether LME crack initiation could happen at lower stress and without high plastic strain in the liquid Pb of the specific oxygen content. In static tests, the stress conditions were pre-set at room temperature before immersion in liquid Pb. It is likely that the stress was slightly relaxed and strain rate diminished in test time. Therefore, no mechanical driving force to fail the oxygen layer was acting. On the other hand, a significant strain rate effect was observed in CERT testing. Applying the higher test rates R0 and R1, much more oxide cracks initiated than in the test with R2. It is likely that the lower strain rates can explain the lower number of surface cracks. However, these were only the oxide cracks and no cracks were found to continue into the bulk of the metal. No tendency of the steel under oxide to damage was observed.

In our previous works [13,14], LME initiation was observed in connection to dynamic strain ageing (DSA) and dislocation creep in the T91 steel. DSA and acceleration of the dislocation creep onset at lower temperatures were both stimulated by slow strain rates. At 400 °C, in Pb, serrations were observed with the applied slow strain rate 1 × 10^−6^ s^−1^, which was identical to the R2 rate of straining of the minimal cross section of the tapered specimen. LME was observed on the notched tensile specimens loaded at R1 and R2.

In this case, the oscillations indicating the serrations appeared on the curves of the tapered specimens in air as well as in liquid Pb (Figure 5), when the slowest test rate R2 was applied. Moreover, the maximum stress increased with decreasing test rate (R0, R1, R2) by 3%–7% in air and by about 4% in Pb. These facts indicate that the T91 testing was also performed within DSA activity. The observed higher elongations at the maximum stress and lower stresses of all three specimens tested in Pb, in comparison to those in air, indicate that the dislocation creeping mechanism was also activated. Despite DSA, creep, and lower resistance of the oxides to the loading, no typical LME cleavage-like cracks initiating from the surface of the tapered specimens were observed. Only the cracking of the oxide layer in sites over threshold stress and strain conditions was observed. These threshold values were evaluated and were about 520 MPa and 2% plastic strain in the case of R0 and R1 test rates, that is, with estimated strain rates from 2 × 10^−5^ to 4 × 10^−3^ s^−1^, but 530 MPa and 2.7% plastic strain in R2 (2–3 × 10^−7^ s^−1^). 

### 4.2. LME Crack Initiation Condition

The experiments presented in this paper did not show any LME crack initiation, since test conditions were selected outside of the LME sensitivity area. However, it is worth adding the new data to the existing results of previous experiments [13,14] in order to reach more general conclusions about LME occurrence in the T91/liquid Pb system. Figure 12 shows an attempt to summarize experimental conditions, at which LME crack initiation in T91 in liquid Pb was studied, including this work. Figure 12a shows the maximum achieved tensile stresses versus the maximum achieved tensile plastic strains plots. It is clear that the stress–strain conditions of the notched specimens where LBE was observed were the highest among all tested specimens. Figure 12b illustrates the dependence of maximum achieved tensile stresses on the applied strain rates. Open symbols stand for the test conditions where LME initiation occurred, while solid ones stand for the tests without LME indication. It is now clear that LME testing was performed under a wide range of conditions allowing preliminary conclusions to be made. 

According to the described data, it can be concluded that LME might occur in the T91 steel/liquid lead system under stresses above 680 MPa and corresponding plastic strain of more than 5% applied with a strain rate of about 1–100 × 10^−6^ s^−1^.

## 5. Conclusions

The effect of the applied stress on T91 performance in liquid lead at 400 °C was studied by static and quasi-static tests. The experiments presented in the paper did not show any LME crack initiation because they were performed in conditions which appeared to be unsuitable for LME occurrence in T91 steel. Specifically:Despite the different oxygen contents and exposure times, during static and quasi-static (CERT) experiments, the double-layer oxide was developed. The oxide thickness values in all specimens were in a range from 1.4 m up to 5 m, in localized areas after the long-term exposure.The load level applied in a static mode (up to 110% of yield strength) was not sufficient to damage the oxide layer.The load level applied in CERT tests with applied strain rates from 2 × 10^−5^ to 4 × 10^−3^s^−1^ was sufficient to break only the oxide layers built at liquid Pb with 1 × 10^−6^ wt. % oxygen, however the liquid lead did not reach the bare metal.No effect of the ground and polished surface was observed.

Based on previous experience and experimental results, LME crack initiation of T91 in liquid Pb with around 1 × 10^−7^ wt. % oxygen can be considered as follows:Temperatures 350–400 °CStress above 680 MPa, plastic strain more than 5%, and strain rate about 1–100 × 10^−6^ s^−1^.

## Figures and Tables

**Figure 1 materials-11-02512-f001:**
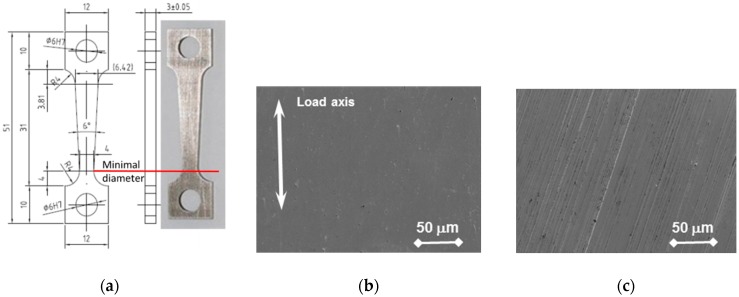
Tapered specimen: (**a**) Drawing, dimensions, and an image with a minimal diameter mark; (**b**) Polished (Ra = 0.005 µm) and (**c**) ground (Ra = 0.047 µm) surfaces before testing. Ra: arithmetical mean roughness.

**Figure 2 materials-11-02512-f002:**
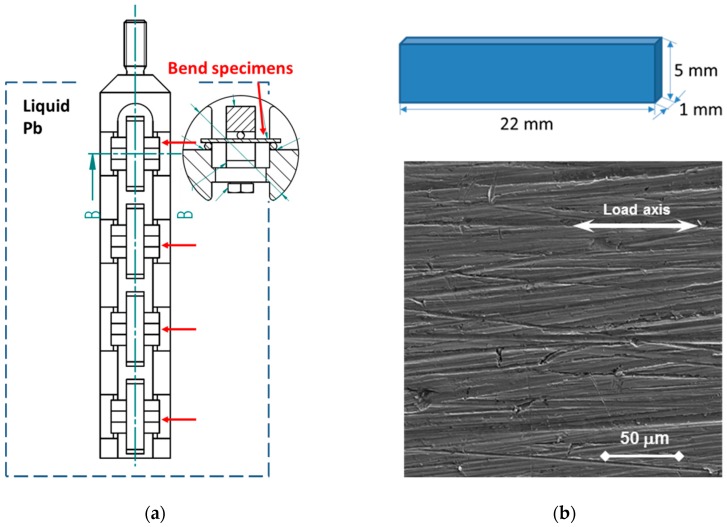
Bend specimen testing: (**a**) Scheme of 4 × 3 specimen holder; red arrows mark sets of three parallel specimens, each set is pre-loaded to a different stress level. (**b**) Schematic and micrograph of the original surface of the bend specimen (Ra = 0.247 µm) before pre-loading and exposure.

**Figure 3 materials-11-02512-f003:**
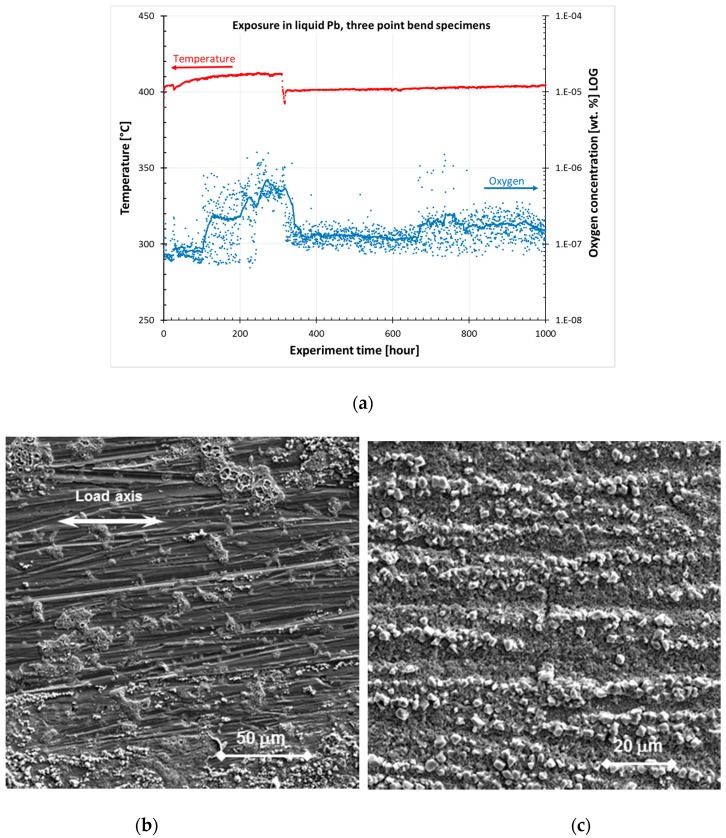
Bend specimens’ exposure in liquid Pb: (**a**) Time dependence of temperature and oxygen content. (**b**,**c**) SEM images of the surface after the 1000 h exposure in Pb and chemical cleaning from Pb showing places with (**b**) non-continuous (T001) and (**c**) continuous oxide layer (T101).

**Figure 4 materials-11-02512-f004:**
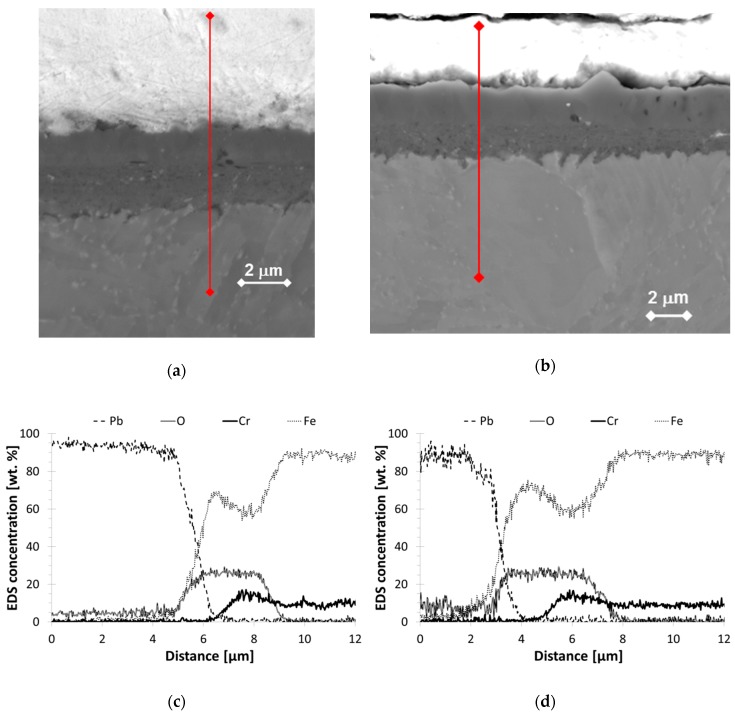
Bend specimens’ exposure to liquid Pb: (**a**) Oxide scale in the site not affected by stress (T110); (**b**) Oxide scale exposed to the stress (100%YS, T100) with red lines marking the position of EDS line scan. (**c**) EDS line scan corresponding to (a) and (**d**) EDS line scan of (b).

**Figure 5 materials-11-02512-f005:**
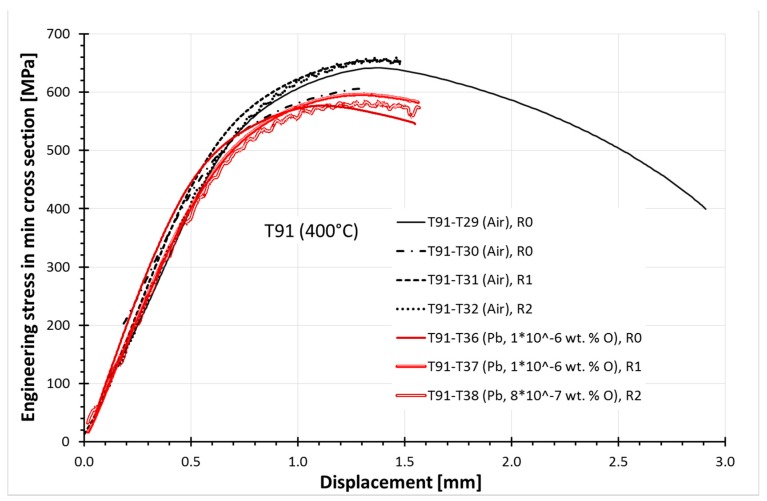
Test results of constant extension rate tensile (CERT) tests of tapered specimens in air and in liquid Pb at 400 °C.

**Figure 6 materials-11-02512-f006:**
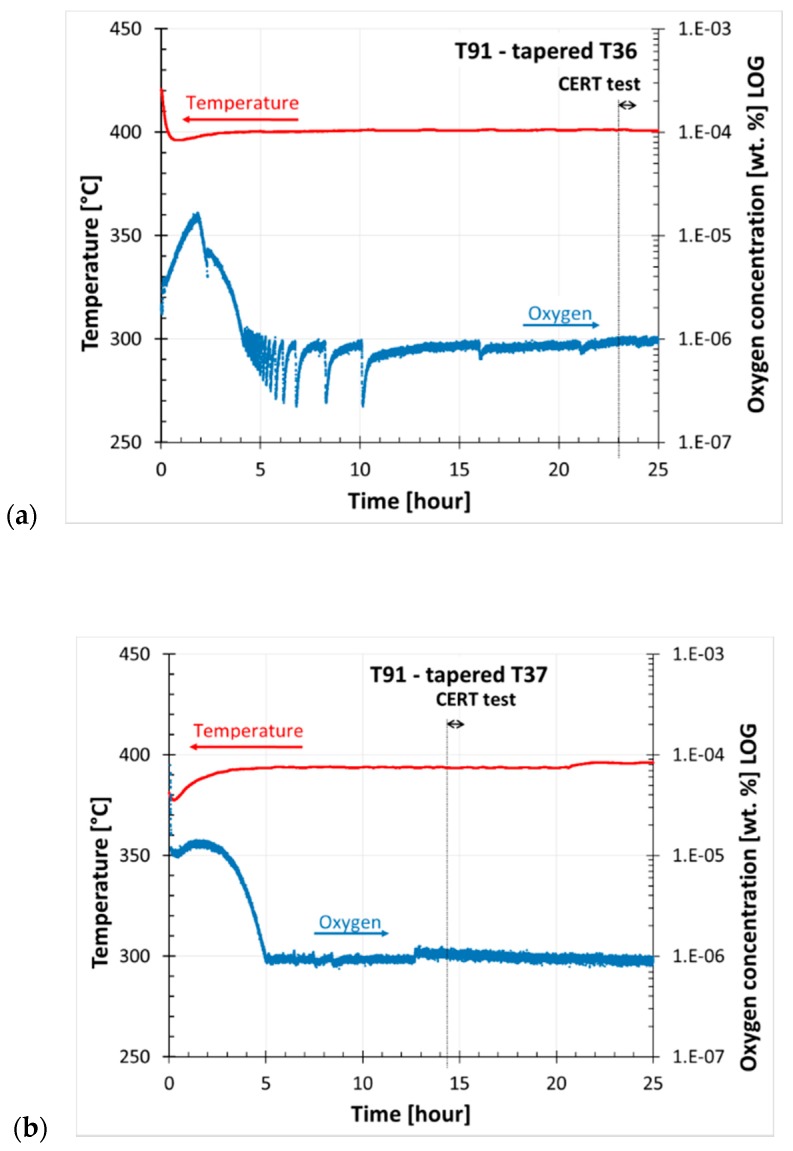
Time dependence of temperature and oxygen concentration in liquid Pb during CERT tests: Tapered specimens (**a**) T36; (**b**) T37; and (**c**) T38. Time started from the filling of the second (test) tank with Pb. The vertical line marks the CERT test start and the arrow marks the test duration.

**Figure 7 materials-11-02512-f007:**
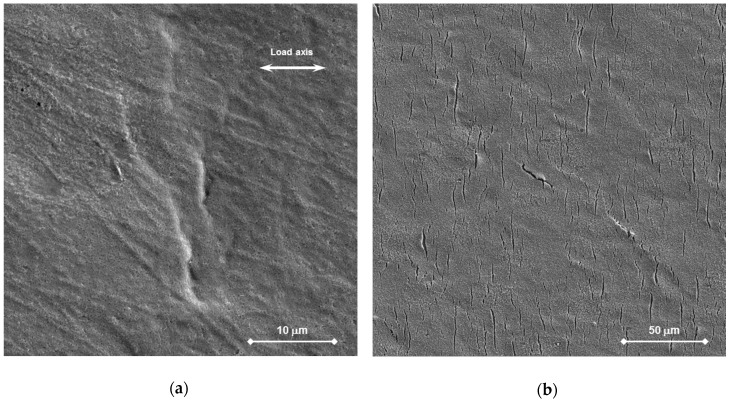
Tapered specimen after CERT by R1 test rate at 400 °C, observation of the polished surface: (**a**) In air, the site of maximum deformation in the minimum cross section showing a layer of oxides obscuring slip bands (T31); (**b**) In Pb and after chemical cleaning, the surface in the minimum cross section showing superficial cracks and indications of deformation below oxides (T37).

**Figure 8 materials-11-02512-f008:**
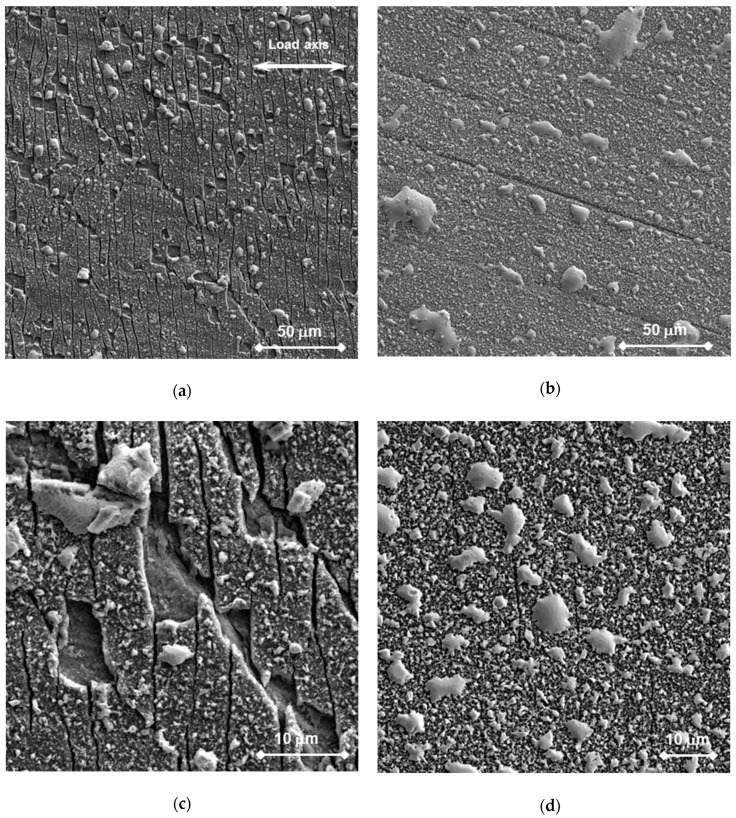
Tapered specimen T36 (R0) in Pb without chemical cleaning: the ground surface (**a**) in the minimum cross section showing a number of cracks in the oxide; (**b**) 5.8 mm from the minimum cross section showing visible machining grooves under the oxide layer; (**c**) detail of (**a**); (**d**) the last (furthest) cracks in the oxide.

**Figure 9 materials-11-02512-f009:**
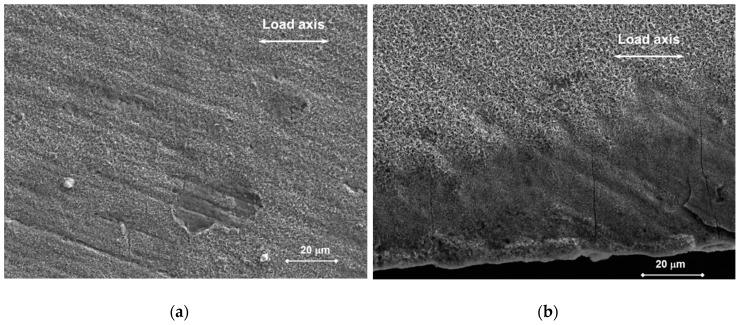
T38 (R2) in Pb, chemical cleaning: (**a**) ground and (**b**) polished surfaces in the minimum cross section.

**Figure 10 materials-11-02512-f010:**
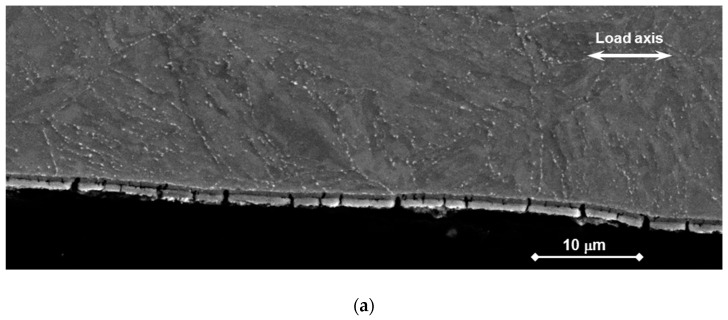
Tapered specimen T36 (R0) after testing in liquid Pb (without chemical cleaning): (**a**) Cross section showing the cracks limited to the oxide (secondary electron (SE)); (**b**) Cracking in the oxide scale going across and between the inner and the outer layers (back-scattering electron (BSE)); (**c**) EDS line scan across the oxide scale.

**Figure 11 materials-11-02512-f011:**
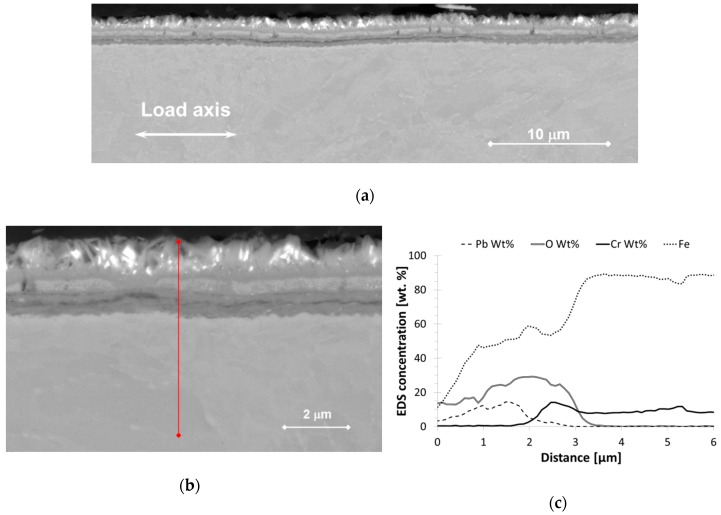
Tapered specimen T38 (R2) after testing in liquid Pb and chemical cleaning: (**a**) Cross section showing the oxide on the ground surface outside the minimum (BSE); (**b)** detail of (**a**); (**c**) EDS line scan across the oxide scale.

**Figure 12 materials-11-02512-f012:**
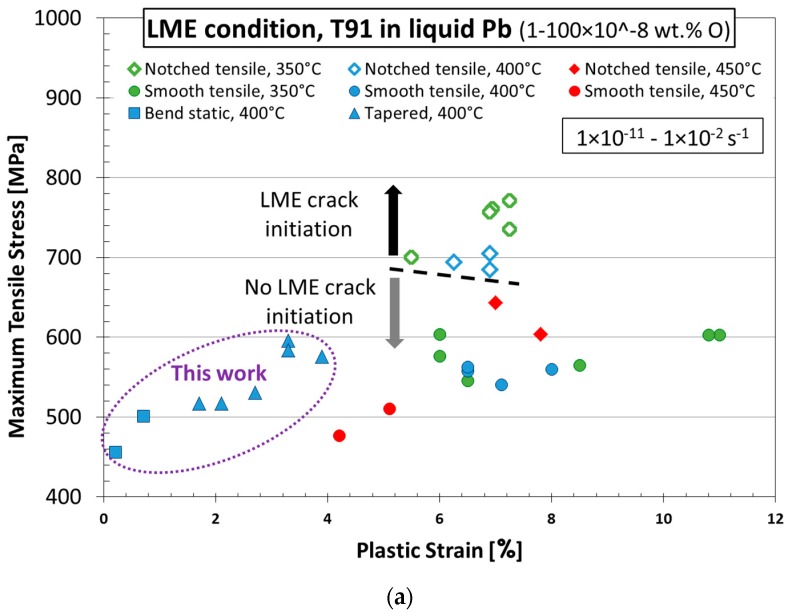
Summary test condition of T91 in liquid Pb mapping LME occurrence. Maximum applied tensile stress in relation to (**a**) maximum total plastic strain and (**b**) strain rates: Data of CERT and static exposure tests with four specimen types (smooth and notched tensile, tapered, and three-point bend) at three temperatures showing the LME area of occurrence [13,14]. Note that LME appeared only in notched tensile specimens at 350 and 400 °C if extremely high stress, over Rm, was applied. This work’s results are highlighted by the dashed-line circle.

**Table 1 materials-11-02512-t001:** The composition of the steel T91 as reported by the producer (wt. %).

Fe	C	Cr	Mo	Mn	Si	Ni	V	Cu	Nb	P	Al	Ti	S	N
Bal.	0.102	8.895	0.889	0.401	0.235	0.121	0.202	0.080	0.079	0.019	0.010	0.004	0.0007	0.048

**Table 2 materials-11-02512-t002:** Mechanical properties of the test material, where Rp0.2 is the yield strength, Rm is the ultimate tensile strength, Am is the deformation at maximum load, A is the fracture deformation, and Z is the reduction of area.

T (°C)	Rp0.2 (MPa)	Rm (MPa)	Am (%)	A (%)	Z (%)
25	575	718	6.7	22	71
400	456	562	2.6	15	-

**Table 3 materials-11-02512-t003:** Results of CERT testing with tapered specimens at 400 °C in air and liquid Pb. f_pl_: the plastic part of total displacement; σ_max_: the maximum stress in the minimum cross section (S_min_); ε_max_: the plastic strain at S_min_; L_x_: the last crack distance; ε_Lx_: the plastic strain in L_x_; σ_th_: the threshold stress; LME: liquid metal embrittlement. “n. a.” means not applicable, “Y” is yes, “N” is no.

No.	Air/Pb	Oxygen (wt. %)	Test Rate (m/s)	Test Stop at	f_pl_ (mm)	ε_max_ (%)	σ_max_ (MPa)	L_x_ (mm)	ε_Lx_ ^1^ (%)	σ_th_ ^2^ (MPa)	LME Y/N
T29	Air	n. a.	R0: 2 × 10^−4^	rupture	2.40	9.2	642	n. a.	n. a.	n. a.	n. a.
T30	Air	n. a.	R0: 2 × 10^−4^	max.	0.60	2.3	606	n. a.	n. a.	n. a.	n. a.
T31	Air	n. a.	R1: 2 × 10^−6^	max.	0.76	2.9	655	n. a.	n. a.	n. a.	n. a.
T32	Air	n. a.	R2: 2 × 10^−8^	max.	0.66	2.5	660	n. a.	n. a.	n. a.	n. a.
T36	Pb	1 × 10^−6^	R0: 2 × 10^−4^	max.	1.02	3.9	577	4.3	2.1	518	N
T37	Pb	1 × 10^−6^	R1: 2 × 10^−6^	max.	0.85	3.3	597	4.3	1.7	517	N
T38	Pb	8 × 10^−7^	R2: 2 × 10^−8^	max.	0.87	3.3	584	4.5	2.7 ^3^	531	N

^1^ Assumed linear strain distribution along the gauge length; ^2^ Threshold stress for cracks on the ground surface; ^3^ Only a few cracks observed.

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
