# Peer review of "Effect of Applied Stress on T91 Steel Performance in Liquid Lead at 400 °C"

_materials, 2018, doi:10.3390/ma11122512_

Reviewer 1 Report

The work presents interesting topics related to T91 steel performance in liquid lead with respect to liquid metal embrittlement (LME). The study has an impact in terms of practical application of T91 steel to development of the lead-cooled fast reactor. However, the reviewer questions mainly the focus of this work. Some shortcomings in the manuscript are discussed below.

General:

Based on the title, abstract and introduction of the manuscript, the objectives are (1) to study the effect of the applied stresses on the mechanical performance of T91 performance in liquid lead and (2) to evaluate threshold stress and strain for the LME initiation. Throughout the paper, I am unsure if this study was well performed in connection to the research objectives. A large portion of the study is based on the oxide and microstructural analysis (figures 3, 6, 7, 8, 9, 10 and 11), indicating that the focus was more on the effect of the oxide structure and properties rather than that of the applied load. Some main conclusions are from literature (figures 12 and 13) rather than from the experimental results of this study, which weakens the originality of this work. Typically, I am not certain if figure 12 is necessary. The conclusions ended up stating that LME did not occur in all cases. While the work contains some interesting, publishable results, the authors should make further efforts to emphasize the originality and novelty of the work. In addition, the authors are requested to reorganize the structure of the manuscript in terms of the focus of the study with some significant thoughts.

Title:

(1) I recommend the authors to avoid specific nomenclature of materials such as T91 in title and abstract. It may be preferable to use "T91 steel" instead of "T91".

-Introduction

(2) The authors stated, “Although one of the HLM coolant operation strategies is to maintain specific oxygen content in liquid metals, in order to build protective oxides on the steels, these media (Pb and LBE) do not have the same effect on materials…”. It would be preferable to add a few more sentences about the needs to precisely control the oxygen content in the industrial practice. In addition, please clarify the logic behind selecting the target oxygen concentration (~10-7 wt. %).

 -2. Materials and methods

(3) Section 2.2: the mechanical testing was performed based on the bend testing and tensile testing of the tapered specimen. It may be helpful to explain the purpose of the use of tapered specimens rather than normal smooth-sided tensile specimen.

(4) Section 2.3.2: it is stated, “Particularly 0, 80, 100 and 110 % Rp0.2 pre-stress levels were applied.”. Meaning of “Rp 0.2 pre-stress level” is not clear (likely, room temperature yield strength of the material?). It is also not clear how the authors calculated the percentage of the load. I assume the authors used the equations in ISO7539-2 to calculate the approximate stress at the mid-point of the convex surface. If so, the equations used should be provided. The authors are requested to describe the pre-bending conditions in more detail (location of the stress measurement or calculation and degree of the bending for each condition).

-3. Results

 (5) Figure 3: it is stated, “…with (b) non-continuous (T001) and (c) continuous oxide layer (T101).”. Please define T001 and T101 in the text. Additionally, are figures 3(b) and (c) taken from different specimens? Same issue in the caption of figure 4.

 (6) Section 3.2, page 7: It is stated, “The threshold stress, σth, is the maximum stress which was achieved during the test, at the position of the last observed crack (Lx).”. The definition of the “last observed crack” should be described when the term is first used in the text. I noticed that the definition of the “last observed crack” is stated in the later sections i.e. the section 3.2.1. Besides, although I understand the intention of the authors, the terminology, “last observed” seems not proper. Moreover, I could not follow the logic or meaning with measuring the threshold stress, σth, at the position of the last observed crack (Lx). There is a similar issue with the measurement of the plastic strain. Please justify the authors’ thoughts.

(7) Captions of figures 8 and 9: it is stated, “without chemical cleaning: the ground surface (a) in the minimum showing many cracks in oxide”. The expression “minimum” is unclear.

-4. Discussion

(8) Section 4.1: it is stated “The oxygen concentration was reduced fast (2 - 5h) and reached the target level before the CERT tests, but the resistance to loading of the developing oxide layer was affected.”. I am not certain if the statement, i.e. the resistance of the oxide to loading was affected by the different oxygen content, is true. The cracking was more likely associated with the fact that for the static bend testing, loading was applied before the immersion in the liquid lead. In this case, oxides may form on the bent specimen and there might be not much stress/strain applied on the oxide formed on the material substrate. In contrast, in the case of the quasi-static experiments, the loading was applied after the immersion, i.e. after the oxide formation. The properties or resistance of the oxide should not be changed if the oxide chemistry is similar between the conditions.

(9) Section 4.1: it is stated “There are likely two reasons why the LME initiation did not appear:

At first… Moreover, small protuberances were observed at the bulk-oxide interface towards the bulk (Fig.4b). These small extensions of the spinel oxide towards the bulk material might be connected to the presence of stresses and the subsequent formation of more numerous inner easy diffusion paths. This might also indicate the beginning stage of development of the internal oxidation zone (IOZ) [19]. These locally accelerated oxidation areas might be at risk of cracking in long-term exposures. They could become local micro notches from which LME could initiate if the high load (higher than 1.1 of the yield strength, i.e. out of operational conditions) would be applied…” I suggest the authors to modify the text to clarify the logic.

(10) Section 4.2: I suggest the authors to consider heavy revision of this section. The paragraph describing figure 12 seems unnecessary in terms of flow of the paper. I do not see any specific reasons for the authors to include the figures which are based entirely on the previous works. 

(11) Figure 13: I am uncertain if there is meaning in comparing the tensile testing results with bend test results in that the loading conditions should be significantly different.

-Conclusions

(12) It is stated, “…Oxidation occurred at the early stage of exposure to liquid Pb and slow loading or applied load for 1000 h did not result in damage to direct contact the T91 steel and the liquid metal. The level of load applied in static mode was not sufficient to damage the oxide layer. The level of load applied in CERT tests was sufficient to break the oxide, but the liquid lead did not reach the steel…”. Once more, the focus of the paper has given to the oxide effects on the LME crack initiation. Furthermore, based on the experimental results shown, it is unclear if the absence of the LME crack is actually due to oxide layer. Are the authors expecting that LME would have occurred in the conditions with lower oxygen content in the liquid lead?

(13) The following statements in the text are not clear:

“Besides some imperfection of the oxide layer (Figure 3c), no clear initiated cracks were observed.”

“It shows that oxidation was accelerated in the areas which might correspond to microstructural elements, the size of about 0.2-0.5 μm, i.e. much smaller than the grain size.”

“…and showed its protectiveness against slow loading up to threshold stress &strain conditions. These were evaluated to be about 520 MPa & 2% plastic strain.”

Author Response

The authors are very grateful to both reviewers for their helpful questions. The answers are included into the reviewer texts below and marked by blue.

Review 1

The work presents interesting topics related to T91 steel performance in liquid lead with respect to liquid metal embrittlement (LME). The study has an impact in terms of practical application of T91 steel to development of the lead-cooled fast reactor. However, the reviewer questions mainly the focus of this work. Some shortcomings in the manuscript are discussed below.

General:

Based on the title, abstract and introduction of the manuscript, the objectives are (1) to study the effect of the applied stresses on the mechanical performance of T91 performance in liquid lead and (2) to evaluate threshold stress and strain for the LME initiation. Throughout the paper, I am unsure if this study was well performed in connection to the research objectives. A large portion of the study is based on the oxide and microstructural analysis (figures 3, 6, 7, 8, 9, 10 and 11), indicating that the focus was more on the effect of the oxide structure and properties rather than that of the applied load. Some main conclusions are from literature (figures 12 and 13) rather than from the experimental results of this study, which weakens the originality of this work. Typically, I am not

certain if figure 12 is necessary. The conclusions ended up stating that LME did not occur in all cases. While the work contains some interesting, publishable results, the authors should make further efforts to emphasize the originality and novelty of the work. In addition, the authors are requested to reorganize the structure of the manuscript in terms of the focus of the study with some significant thoughts.

Authors’ response: Besides major English revision the introduction, discussion and the conclusions were revised to more focus on the effect of applied stress and staining conditions. We believe that now the text is more consistent.

Title:

(1) I recommend the authors to avoid specific nomenclature of materials such as T91 in title and abstract. It may be preferable to use "T91 steel" instead of "T91".

Authors’ response: Your suggestion was applied.

-Introduction

(2) The authors stated, “Although one of the HLM coolant operation strategies is to maintain specific oxygen content in liquid metals, in order to build protective oxides on the steels, these media (Pb and LBE) do not have the same effect on materials…”. It would be preferable to add a few more sentences about the needs to precisely control the oxygen content in the industrial practice. In addition, please clarify the logic behind selecting the target oxygen concentration (~10-7 wt. %).

Authors’ response: Thank for your suggestion, the text is improved.

 -2. Materials and methods

(3) Section 2.2: the mechanical testing was performed based on the bend testing and tensile testing of the tapered specimen. It may be helpful to explain the purpose of the use of tapered specimens rather than normal smooth-sided tensile specimen.

Authors’ response:  Based on your suggestion the new text “The 3° taper creates variation of stress and strain along the gauge length. Maximum stress is always in the minimum cross-section; the stress close to the end of the in the wider part stays elastic and does not overcome the yield strength. It allows the identification of threshold stress and strain condition for the crack initiation within a single test.” is added.

(4) Section 2.3.2: it is stated, “Particularly 0, 80, 100 and 110 % Rp0.2 pre-stress levels were applied.”. Meaning of “Rp 0.2 pre-stress level” is not clear (likely, room temperature yield strength of the material?). It is also not clear how the authors calculated the percentage of the load. I assume the authors used the equations in ISO7539-2 to calculate the approximate stress at the mid-point of the convex surface. If so, the equations used should be provided. The authors are requested to describe the pre-bending conditions in more detail (location of the stress measurement or calculation and degree of the bending for each condition).

 Authors’ response: The text is now more detailed.

-3. Results

 (5) Figure 3: it is stated, “…with (b) non-continuous (T001) and (c) continuous oxide layer (T101).”. Please define T001 and T101 in the text. Additionally, are figures 3(b) and (c) taken from different specimens? Same issue in the caption of figure 4.

Authors’ response: Yes, the names in the bracket are the specimen’s names. Now, these names are defined in the text of the chapter “Specimens”.

 (6) Section 3.2, page 7: It is stated, “The threshold stress, σth, is the maximum stress which was achieved during the test, at the position of the last observed crack (Lx).”. The definition of the “last observed crack” should be described when the term is first used in the text. I noticed that the definition of the “last observed crack” is stated in the later sections i.e. the section 3.2.1. Besides, although I understand the intention of the authors, the terminology, “last observed” seems not proper. Moreover, I could not follow the logic or meaning with measuring the threshold stress, σth, at the position of the last observed crack (Lx). There is a similar issue with the measurement of the plastic strain. Please justify the authors’ thoughts.

Authors’ response: More detail explaining text is included in 3.2.1.

(7) Captions of figures 8 and 9: it is stated, “without chemical cleaning: the ground surface (a) in the minimum showing many cracks in oxide”. The expression “minimum” is unclear.

 Authors’ response: More text was added into the captures. The minimum means the minimal cross-section of the tapered specimen.

-4. Discussion

(8) Section 4.1: it is stated “The oxygen concentration was reduced fast (2 - 5h) and reached the target level before the CERT tests, but the resistance to loading of the developing oxide layer was affected.”. I am not certain if the statement, i.e. the resistance of the oxide to loading was affected by the different oxygen content, is true. The cracking was more likely associated with the fact that for the static bend testing, loading was applied before the immersion in the liquid lead. In this case, oxides may form on the bent specimen and there might be not much stress/strain applied on the oxide formed on the material substrate. In contrast, in the case of the quasi-static experiments, the loading was applied after the immersion, i.e. after the oxide formation. The properties or resistance of the oxide should not be changed if the oxide chemistry is similar between the conditions.

Authors’ response: Based on your comments, the whole discussion has been rephrased. We agree with you that the oxides should show some differences if resistance was different. However, by the applied SEM and EDS resolutions we are not able to find it. Further work (FIB plus TEM) is necessary.

 (9) Section 4.1: it is stated “There are likely two reasons why the LME initiation did not appear:

At first… Moreover, small protuberances were observed at the bulk-oxide interface towards the bulk (Fig.4b). These small extensions of the spinel oxide towards the bulk material might be connected to the presence of stresses and the subsequent formation of more numerous inner easy diffusion paths. This might also indicate the beginning stage of development of the internal oxidation zone (IOZ) [19]. These locally accelerated oxidation areas might be at risk of cracking in long-term exposures. They could become local micro notches from which LME could initiate if the high load (higher than 1.1 of the yield strength, i.e. out of operational conditions) would be applied…” I suggest the authors to modify the text to clarify the logic.

 Authors’ response: Yes, the text is now modified.

(10) Section 4.2: I suggest the authors to consider heavy revision of this section. The paragraph describing figure 12 seems unnecessary in terms of flow of the paper. I do not see any specific reasons for the authors to include the figures which are based entirely on the previous works. 

Authors’ response: Your suggestion was applied and the Figure 12 as well as the text about it was removed. Yes, it was repetition of what was said in introduction.

(11) Figure 13: I am uncertain if there is meaning in comparing the tensile testing results with bend test results in that the loading conditions should be significantly different.

Authors’ response: Maybe now it is clearer when we add one more graph (Fig 12b) mapping LME occurrence versus strain rate. These tests form a lower boundary of the strain rates.

 -Conclusions

(12) It is stated, “…Oxidation occurred at the early stage of exposure to liquid Pb and slow loading or applied load for 1000 h did not result in damage to direct contact the T91 steel and the liquid metal. The level of load applied in static mode was not sufficient to damage the oxide layer. The level of load applied in CERT tests was sufficient to break the oxide, but the liquid lead did not reach the steel…”. Once more, the focus of the paper has given to the oxide effects on the LME crack initiation. Furthermore, based on the experimental results shown, it is unclear if the absence of the LME crack is actually due to oxide layer. Are the authors expecting that LME would have occurred in the conditions with lower oxygen content in the liquid lead?

Authors’ response: Yes, we agree that it is not only oxides effect, but also strain rate effect which affected the observed cracking. However, we have also other results as well as more experience in LBE environment indicating that LME can initiate in the Pb with lower oxygen content out of the stable oxide window. If the oxygen is so low, that dissolution takes place, LME initiation is highly likely.

(13) The following statements in the text are not clear:

“Besides some imperfection of the oxide layer (Figure 3c), no clear initiated cracks were observed.”

 “It shows that oxidation was accelerated in the areas which might correspond to microstructural elements, the size of about 0.2-0.5 μm, i.e. much smaller than the grain size.”

“…and showed its protectiveness against slow loading up to threshold stress &strain conditions. These were evaluated to be about 520 MPa & 2% plastic strain.”

Authors’ response: These texts were improved in frame of the English revision.

Reviewer 2 Report

F/M steels are a group of important candidate materials for liquid metal cooled Gen IV reactor and SMR design concepts. It has been considered as fuel cladding materials, heat exchanger tubes in SFR, structure materials in TWR. The coolant compatibility is always one of the most critical issues for LMFR. The current study is targeted to address the liquid metal induced stress corrosion cracking of T91 steel for potential LFR application, which is the key evaluation for practical application. However, there are some key issues need to be clarify before the current manuscript can be accepted. The detailed comments are listed as follows:

The design of tapered samples need to be justified. Typically, a tapered sample is designed with the thin section at max tensile stress and thick section at yield strength and therefore, one tapered samples can produce multiple data points for cracking. This doesn't seem the case for the current study and need some clarification.

Please clarify why CERT was chosen for tapered sample rather than constant load case. The reviewer feels that constant load mode will have better defined stress condition at a given point than CERT test, especially for tapered sample.

The testing condition of the materials was actually at a less interesting region, especially consider the fact that the cracking boundaries has not been fully established. It will be great that future work can clarify this point.

Suggest to add strain rate in Table 3, which is much more important than test rate.

Please justify the current sample finish condition vs. electropolishing condition. 

Consider to clarify the hardness measurement parameters, applied load and number of indents.

A similar Figure to Figure 13 is strongly suggested with the x-axis of strain rate and y-axis of stress level. This figure is more convincing than the current one in defining the cracking initiation boundary.

If the notched tensile sample data was removed, where is the cracking initiation boundaries? This indicates that the boundaries was determined by only one data point. The reviewer believes that solid data are still needed to clarify the cracking initiation boundary. The conclusion related to stress corrosion cracking condition is therefore not convincing.

The T91 steels used here doesn't seem to be fully tempered, where the strength was higher than the typical T91 and with a much lower ductility [M.Song et al, Acta, 2016]. Did the cracking initiation boundary define by the same heat treatment schedule?

Author Response

Review 2

F/M steels are a group of important candidate materials for liquid metal cooled Gen IV reactor and SMR design concepts. It has been considered as fuel cladding materials, heat exchanger tubes in SFR, structure materials in TWR. The coolant compatibility is always one of the most critical issues for LMFR. The current study is targeted to address the liquid metal induced stress corrosion cracking of T91 steel for potential LFR application, which is the key evaluation for practical application. However, there are some key issues need to be clarify before the current manuscript can be accepted. The detailed comments are listed as follows:

1. The design of tapered samples need to be justified. Typically, a tapered sample is designed with the thin section at max tensile stress and thick section at yield strength and therefore, one tapered samples can produce multiple data points for cracking. This doesn't seem the case for the current study and need some clarification.

Authors’ response: Of course that, it is the case, too. When loaded, the maximum stress is always in the minimum cross-section and the wider end stays all time under elastic loading. We added more explaining text in Line 83-86.

2. Please clarify why CERT was chosen for tapered sample rather than constant load case. The reviewer feels that constant load mode will have better defined stress condition at a given point than CERT test, especially for tapered sample.

Authors’ response: The tapered shape allows testing wide range of stresses. We added more explaining text in Line 83-86.

3. The testing condition of the materials was actually at a less interesting region, especially consider the fact that the cracking boundaries has not been fully established. It will be great that future work can clarify this point.

Authors’ response: The testing was aimed to more-less simulate a steady-state operation of LFR. We did the tests because no similar results are available. The usual LME testing is in conditions very much over a future operation condition.

4. Suggest to add strain rate in Table 3, which is much more important than test rate.

Authors’ response: In the Table, there is only test rate because it is valid for the whole specimen. Strain rate is different for different cross-sections.

5. Please justify the current sample finish condition vs. electropolishing condition.

Authors’ response: We used only fine sand paper polishing without chemicals. This way the nanostructure under machined surface was not completely removed but minimized. When using electro-polishing, the layer can be removed totally.

6. Consider to clarify the hardness measurement parameters, applied load and number of indents.

Authors’ response: Standard automatic procedure for the hardness measurement was applied. We listed the hardness as one of parameters characterizing the surface. It was not aimed as investigation.

7. A similar Figure to Figure 13 is strongly suggested with the x-axis of strain rate and y-axis of stress level. This figure is more convincing than the current one in defining the cracking initiation boundary.

Authors’ response: Thanks a lot for the excellent idea. The graph according your suggestion was added as Figure 12b. We would like to keep the first one, too. It is now Figure 12a.

8. If the notched tensile sample data was removed, where is the cracking initiation boundaries? This indicates that the boundaries was determined by only one data point. The reviewer believes that solid data are still needed to clarify the cracking initiation boundary. The conclusion related to stress corrosion cracking condition is therefore not convincing.

Authors’ response: Please see our response to 3. Not found LME in these test condition is a good sign for designers of future systems. They not need to know what condition LME occurs, but if it will occur in their proposed condition.

9. The T91 steels used here doesn't seem to be fully tempered, where the strength was higher than the typical T91 and with a much lower ductility [M.Song et al, Acta, 2016]. Did the cracking initiation boundary define by the same heat treatment schedule?

Authors’ response: Yes, the present data as well as the previous one were done for the same heat batch of T91 steel.

Round  2

Reviewer 1 Report

The authors' answer and their thoughts are reasonable. The manuscript has been improved significantly. I do not have further comments.

Reviewer 2 Report

The reviewer is happy with the current format of the paper.